# Logarithmic Pruning is All You Need

**Laurent Orseau**
DeepMind, London, UK
lorseau@google.com

**Marcus Hutter**
DeepMind, London, UK
www.hutter1.net

**Omar Rivasplata**
DeepMind, London, UK
rivasplata@google.com

## Abstract

The Lottery Ticket Hypothesis is a conjecture that every large neural network contains a subnetwork that, when trained in isolation, achieves comparable performance to the large network. An even stronger conjecture has been proven recently: Every sufficiently overparameterized network contains a subnetwork that, at random initialization, but without training, achieves comparable accuracy to the trained large network. This latter result, however, relies on a number of strong assumptions and guarantees a polynomial factor on the size of the large network compared to the target function. In this work, we remove the most limiting assumptions of this previous work while providing significantly tighter bounds: the overparameterized network only needs a logarithmic factor (in all variables but depth) number of neurons per weight of the target subnetwork.

## 1   Introduction

The recent success of neural network (NN) models in a variety of tasks, ranging from vision [Khan et al., 2020] to speech synthesis [van den Oord et al., 2016] to playing games [Schrittwieser et al., 2019, Ebendt and Drechsler, 2009], has sparked a number of works aiming to understand how and why they work so well. Proving theoretical properties for neural networks is quite a difficult task, with challenges due to the intricate composition of the functions they implement and the high-dimensional regimes of their training dynamics. The field is vibrant but still in its infancy, many theoretical tools are yet to be built to provide guarantees on what and how NNs can learn. A lot of progress has been made towards understanding the convergence properties of NNs (see *e.g.*, Allen-Zhu et al. [2019], Zou and Gu [2019] and references therein). The fact remains that training and deploying deep NNs has a large cost [Livni et al., 2014], which is problematic. To avoid this problem, one could stick to a small network size. However, it is becoming evident that there are benefits to using oversized networks, as the literature on overparametrized models [Ma et al., 2018] points out. Another solution, commonly used in practice, is to prune a trained network to reduce the size and hence the cost of prediction/deployment. While missing theoretical guarantees, experimental works show that pruning can considerably reduce the network size without sacrificing accuracy.

The influential work of Frankle and Carbin [2019] has pointed out the following observation: a) train a large network for long enough and observe its performance on a dataset, b) prune it substantially to reveal a much smaller subnetwork with good (or better) performance, c) reset the weights of the subnetwork to their original values and remove the rest of the weights, and d) retrain the subnetwork in isolation; then the subnetwork reaches the same test performance as the large network, and trains faster. Frankle and Carbin [2019] thus conjecture that every successfully trained network contains a much smaller subnetwork that, when trained in isolation, has comparable performance to the large network, without sacrificing computing time. They name this phenomenon the Lottery Ticket Hypothesis, and a 'winning ticket' is a subnetwork of the kind just described.

Ramanujan et al. [2019] went even further by observing that if the network architecture is large enough, then it contains a smaller network that, *even without any training*, has comparable accuracy to the trained large network. They support their claim with empirical results using a new pruning

algorithm, and even provide a simple asymptotic justification that we rephrase here: Starting from the inputs and progressing toward the outputs, for any neuron of the target network, sample as many neurons as required until one calculates a function within small error of the target neuron; then, after pruning the unnecessary neurons, the newly generated network will be within some small error of the target network. Interestingly, Ulyanov et al. [2018] pointed out that randomly initialized but untrained ConvNets already encode a great deal of the image statistics required for restoration tasks such as de-noising and inpainting, and the only prior information needed to do them well seems to be contained in the network structure itself, since no part of the network was learned from data.

Very recently, building upon the work of Ramanujan et al. [2019], Malach et al. [2020] proved a significantly stronger version of the "pruning is all you need" conjecture, moving away from asymptotic results to non-asymptotic ones: With high probability, any target network of $\ell$ layers and $n$ neurons per layer can be approximated within $\varepsilon$ accuracy by pruning a larger network whose size is polynomial in the size of the target network. To prove their bounds, Malach et al. [2020] make assumptions about the norms of the inputs and of the weights. This polynomial bound already tells us that unpruned networks contain many 'winning tickets' even without training. Then it is natural to ask: could the most important task of gradient descent be pruning?

Building on top of these previous works, we aim at providing stronger theoretical guarantees still based on the motto that "pruning is all you need" but hoping to provide further insights into how 'winning tickets' may be found. In this work we relax the aforementioned assumptions while greatly strengthening the theoretical guarantees by improving from polynomial to logarithmic order in all variables except the depth, for the number of samples required to approximate one target weight.

**How this paper is organized.** After some notation (Section 2) and the description of the problem (Section 3), we provide a general approximation propagation lemma (Section 4), which shows the effect of the different variables on the required accuracy. Next, we show how to construct the large, fully-connected ReLU network in Section 5 identical to Malach et al. [2020], except that weights are sampled from a hyperbolic weight distribution instead of a uniform one. We then give our theoretical results in Section 6, showing that only $\tilde{O}(\log(\ell n_{\max}/\varepsilon))$ neurons per target weight are required under some similar conditions as the previous work (with $\ell$ layers, $n_{\max}$ neurons per layer and $\varepsilon$ accuracy) or $\tilde{O}(\ell \log(n_{\max}/\varepsilon))$ (with some other dependencies inside the log) if these conditions are relaxed. For completeness, the most important technical result is included in Section 7. Other technical results, a table of notation, and further ideas can be found in Appendix C.

## 2   Notation and definitions

A network architecture $A(\ell, \mathbf{n}, \boldsymbol{\sigma})$ is described by a positive integer $\ell$ corresponding to the number of fully connected feed-forward layers, and a list of positive integers $\mathbf{n} = (n_0, n_1, \ldots, n_\ell)$ corresponding to the profile of widths, where $n_i$ is the number of neurons in layer $i \in [\ell] = \{1, \ldots, \ell\}$ and $n_0$ is the input dimension, and a list of activation functions $\boldsymbol{\sigma} = (\sigma_1, \ldots, \sigma_\ell)$—all neurons in layer $i$ use the activation function $\sigma_i$. Networks from the architecture $A(\ell, \mathbf{n}, \boldsymbol{\sigma})$ implement functions from $\mathbb{R}^{n_0}$ to $\mathbb{R}^{n_\ell}$ that are obtained by successive compositions: $\mathbb{R}^{n_0} \longrightarrow \mathbb{R}^{n_1} \longrightarrow \cdots \longrightarrow \mathbb{R}^{n_\ell}$.

Let $F$ be a *target network* from architecture $A(\ell, \mathbf{n}, \boldsymbol{\sigma})$. The composition of such $F$ is as follows: Each layer $i \in [\ell]$ has a matrix $W_i^* \in [-w_{\max}, w_{\max}]^{n_i \times n_{i-1}}$ of connection weights, and an activation function $\sigma_i$, such as tanh, the logistic sigmoid, ReLU, Heaviside, etc. The network takes as input a vector $x \in \mathcal{X} \subset \mathbb{R}^{n_0}$ where for example $\mathcal{X} = \{-1, 1\}^{n_0}$ or $\mathcal{X} = [0, x_{\max}]^{n_0}$, etc. In layer $i$, the neuron $j$ with in-coming weights $W_{i,j}^*$ calculates $f_{i,j}(y) = \sigma_i(W_{i,j}^* y)$, where $y \in \mathbb{R}^{n_{i-1}}$ is usually the output of the previous layer. Note that $W_{i,j}^*$ is the $j$-th row of the matrix $W_i^*$. The vector $f_i(y) = [f_{i,1}(y), \ldots, f_{i,n_i}(y)]^\top \in \mathbb{R}^{n_i}$ denotes the output of the whole layer $i$ when it receives $y \in \mathbb{R}^{n_{i-1}}$ from the previous layer. Furthermore, for a given *network input* $x \in \mathcal{X}$ we recursively define $F_i(x)$ by setting $F_0(x) = x$, and for $i \in [l]$ then $F_i(x) = f_i(F_{i-1}(x))$. The output of neuron $j \in [n_i]$ in layer $i$ is $F_{i,j}(x) = f_{i,j}(F_{i-1}(x))$. The *network output* is $F(x) = F_\ell(x)$.

For an activation function $\sigma(.)$, let $\lambda$ be its Lipschitz factor (when it exists), that is, $\lambda$ is the smallest real number such that $|\sigma(x) - \sigma(y)| \leq \lambda|x - y|$ for all $(x, y) \in \mathbb{R}^2$. For ReLU and tanh we have $\lambda = 1$, and for the logistic sigmoid, $\lambda = 1/4$. Let $\lambda_i$ be the $\lambda$ corresponding to the activation function $\sigma_i$ of all the neurons in layer $i$, and let $\lambda_{\max} = \max_{i \in [\ell]} \lambda_i$.

Define $n_{\max} = \max_{i \in [0..\ell]} n_i$ to be the maximum number of neurons per layer. The total number of connection weights in the architecture $A(\ell, \mathbf{n}, \boldsymbol{\sigma})$ is denoted $N^*$, and we have $N^* \leq \ell n_{\max}^2$.

For all $x \in \mathcal{X}$, let $F_{\max}(x) = \max_{i \in [\ell]} \max_{j \in [n_{i-1}]} |F_{i-1,j}(x)|$ be the maximum activation at any layer of a target network $F$, including the network inputs but excluding the network outputs. We also write $F_{\max}(\mathcal{X}) = \sup_{x \in \mathcal{X}} F_{\max}(x)$; when $\mathcal{X}$ is restricted to the set of inputs of interest (not necessarily the set of all possible inputs) such as a particular dataset, we expect $F_{\max}(\mathcal{X})$ to be bounded by a small constant in most if not all cases. For example, $F_{\max}(\mathcal{X}) \leq 1$ for a neural network with only sigmoid activations and inputs in $[-1, 1]^{n_0}$. For ReLU activations, $F_{\max}(\mathcal{X})$ can in principle grow as fast as $(n_{\max} w_{\max})^\ell$, but since networks with sigmoid activations are universal approximators, we expect that for all functions that can be approximated with a sigmoid network there is a ReLU network calculating the same function with $F_{\max}(\mathcal{X}) = O(1)$.

The *large network* $G$ has an architecture $A(\ell', \mathbf{n}', \boldsymbol{\sigma}')$, possibly wider and deeper than the target network $F$. The *pruned network* $\hat{G}$ is obtained by pruning (setting to 0) many weights of the large network $G$. For each layer $i \in [\ell']$, and each pair of neurons $j_1 \in [n_i]$ and $j_2 \in [n_{i-1}]$, for the weight $w_{i,j_1,j_2}$ of the large network $G$ there is a corresponding mask $b_{i,j_1,j_2} \in \{0, 1\}$ such that the weight of the pruned network $\hat{G}$ is $w'_{i,j_1,j_2} = b_{i,j_1,j_2} w_{i,j_1,j_2}$. The pruned network $\hat{G}$ will have a different architecture from $F$, but at a higher level (by grouping some neurons together) it will have the same 'virtual' architecture, with virtual weights $\hat{W}$. As in previous theoretical work, we consider an 'oracle' pruning procedure, as our objective is to understand the limitations of even the best pruning procedures.

For a matrix $M \in [-c, c]^{n \times m}$, we denote by $\|M\|_2$ its spectral norm, equal to its largest singular value, and its max-norm is $\|M\|_{\max} = \max_{i,j} |M_{i,j}|$. In particular, for a vector $v$, we have $\|Mv\|_2 \leq \|M\|_2 \|v\|_2$ and $\|M\|_{\max} \leq \|M\|_2 \leq \sqrt{nm} \|M\|_{\max}$ and also $\|M\|_{\max} \leq c$. This means for example that $\|M\|_2 \leq 1$ is a stronger condition than $\|M\|_{\max} \leq 1$.

## 3 Objective

**Objective:** Given an architecture $A(\ell, \mathbf{n}, \boldsymbol{\sigma})$ and accuracy $\epsilon > 0$, construct a network $G$ from some larger architecture $A(\ell', \mathbf{n}', \boldsymbol{\sigma}')$, such that if the weights of $G$ are randomly initialized *(no training)*, then for any target network $F$ from $A(\ell, \mathbf{n}, \boldsymbol{\sigma})$, setting some of the weights of $G$ to 0 *(pruning)* reveals a subnetwork $\hat{G}$ such that with high probability,

$$\sup_{x \in \mathcal{X}} \|F(x) - \hat{G}(x)\|_2 \leq \varepsilon$$

**Question:** How large must $G$ be to contain all such $\hat{G}$? More precisely, how many more neurons or how many more weights must $G$ have compared to $F$?

Ramanujan et al. [2019] were the first to provide a formal asymptotic argument proving that such a $G$ can indeed exist at all. Malach et al. [2020] went substantially further by providing the first polynomial bound on the size of $G$ compared to the size of the target network $F$. To do so, they make the following assumptions on the target network: (i) the inputs $x \in \mathcal{X}$ must satisfy $\|x\|_2 \leq 1$, and at all layers $i \in [\ell]$: (ii) the weights must be bounded in $[-1/\sqrt{n_{\max}}, 1/\sqrt{n_{\max}}]$, (iii) they must satisfy $\|W_i^*\|_2 \leq 1$ at all layers $i$, and (iv) the number of non-zero weights at layer $i$ must be less than $n_{\max}$: $\|W_i^*\|_0 \leq n_{\max}$. Note that these constraints imply that $F_{\max}(\mathcal{X}) \leq 1$. Then under these conditions, they prove that any ReLU network with $\ell$ layers and $n_{\max}$ neurons per layer can be approximated[1] within $\varepsilon$ accuracy with probability $1 - \delta$ by pruning a network $G$ with $2\ell$ ReLU layers and each added intermediate layer has $n_{\max}^2 \lceil \frac{64\ell^2 n_{\max}^3}{\varepsilon^2} \log \frac{2n_{\max}^2 \ell}{\delta} \rceil$ neurons. These assumptions are rather strong, as in general this forces the activation signal to decrease quickly with the depth. Relaxing these assumptions while using the same proof steps would make the bounds exponential in the number of layers. We build upon the work of Ramanujan et al. [2019], Malach et al. [2020], who gave the first theoretical results on the Lottery Ticket Hypothesis, albeit under restrictive assumptions. Our work re-uses some of their techniques to provide sharper bounds while removing these assumptions.

## 4   Approximation Propagation

In this section, we analyze how the approximation error between two networks with the same architecture propagates through the layers. The following lemma is a generalization of the (end of the) proof of Malach et al. [2020, Theorem A.6] that removes their aforementioned assumptions and provides better insight into the impact of the different variables on the required accuracy, but is not sufficient in itself to obtain better bounds. For two given networks with the same architecture, it determines what accuracy is needed on each individual weight so the outputs of the two neural networks differ by at most $\varepsilon$ on any input. Note that no randomization appears at this stage.

**Lemma 1** (Approximation propagation)**.**  Consider two networks $F$ and $\hat{G}$ with the same architecture $A(\ell, \mathbf{n}, \boldsymbol{\sigma})$ with respective weight matrices $W^*$ and $\hat{W}$, each weight being in $[-w_{\max}, w_{\max}]$. Given $\varepsilon > 0$, if for each weight $w^*$ of $F$ the corresponding weight $\hat{w}$ of $\hat{G}$ we have $|w^* - \hat{w}| \leq \varepsilon_w$, and if

$$\varepsilon_w \leq \varepsilon \left/ \left( e\, \ell\, \lambda_{\max}\, n_{\max}^{3/2}\, F_{\max}(\mathcal{X}) \prod_{i=1}^{\ell} \max\{1, \lambda_i \|\hat{W}_i\|_2\} \right) \right. , \quad \text{then} \quad \sup_{x \in \mathcal{X}} \|F(x) - \hat{G}(x)\|_2 \leq \varepsilon .$$

The proof is given in Appendix C.

**Example 2.**  Consider an architecture with only ReLU activation function ($\lambda = 1$), weights in $[-1, 1]$ and assume that $F_{\max}(\mathcal{X}) = 1$ and take the worst case $\|\hat{W}_i\|_2 \leq w_{\max} n_{\max} = n_{\max}$, then Lemma 1 tells us that the approximation error on each individual weight must be at most $\varepsilon_w \leq \varepsilon/(e\ell n_{\max}^{3/2+\ell})$ so as to guarantee that the approximation error between the two networks is at most $\varepsilon$. This is exponential in the number of layers. If we assume instead that $\|\hat{W}_i\|_2 \leq 1$ as in previous work then this reduces to a mild polynomial dependency: $\varepsilon_w \leq \varepsilon/(e\ell n_{\max}^{3/2})$. $\qquad\qquad\triangle$

## 5   Construction of the ReLU Network $G$ and Main Ideas

We now explain how to construct the large network $G$ given only the architecture $A(\ell, \mathbf{n}, \boldsymbol{\sigma})$, the accuracy $\varepsilon$, and the domain $[-w_{\max}, w_{\max}]$ of the weights. Apart from this, the target network $F$ is unknown. In this section all activation functions are ReLU $\sigma(x) = \max\{0, x\}$, and thus $\lambda = 1$.

We use a similar construction of the large network $G$ as Malach et al. [2020]: both the target network $F$ and the large network $G$ consist of fully connected ReLU layers, but $G$ may be wider and deeper. The weights of $F$ are in $[-w_{\max}, w_{\max}]$. The weights for $G$ (at all layers) are all sampled from the same distribution, the only difference with the previous work is the distribution of the weights: we use a hyperbolic distribution instead of a uniform one.

Between layer $i - 1$ and $i$ of the target architecture, for the large network $G$ we insert an intermediate layer $i - 1/2$ of ReLU neurons. Layer $i - 1$ is fully connected to layer $i - 1/2$ which is fully connected to layer $i$. By contrast to the target network $F$, in $G$ the layers $i - 1$ and $i$ are not directly connected. The insight of Malach et al. [2020] is to use two intermediate (fully connected ReLU) neurons $z^+$ and $z^-$ of the large network $G$ to mimic one weight $w^*$ of the target network (seeFig. 1): Calling $z_{\text{in}}^+, z_{\text{out}}^+, z_{\text{in}}^-, z_{\text{out}}^-$ the input and output weights of $z^+$ and $z^-$ that match the input and output of the connection $w^*$, then in the pruned network $\hat{G}$ all connections apart from these 4 are masked out by a binary mask $\boldsymbol{b}$ set to 0. These two neurons together implement a 'virtual' weight $\hat{w}$ and calculate the function $x \mapsto \hat{w}x$ by taking advantage of the identity $x = \sigma(x) - \sigma(-x)$:

$$\hat{w} = z_{\text{out}}^+ \sigma(z_{\text{in}}^+ x) + z_{\text{out}}^- \sigma(z_{\text{in}}^- x)$$

Hence, if $z_{\text{in}}^+ \approx w^* \approx -z_{\text{in}}^-$ and $z_{\text{out}}^+ \approx 1 \approx -z_{\text{out}}^-$, the virtual weight $\hat{w}$ made of $z^+$ and $z^-$ is approximately $w^*$. Then, for each target weight $w^*$, Malach et al. [2020] sample many such intermediate neurons to ensure that two of them can be pruned so that $|w^* - \hat{w}| \leq \varepsilon_w$ with high probability. This requires $\Omega(1/\varepsilon_w^2)$ samples and, when combined with Lemma 1 (see Example 2), makes the general bound on the whole network grow exponentially in the number of layers, unless strong constraints are imposed.

To obtain a logarithmic dependency on $\varepsilon_w$, we use three new insights that take advantage of the composability of neural networks: 1) 'binary' decomposition of the weights, 2) product weights, and 3) batch sampling. We detail them next.

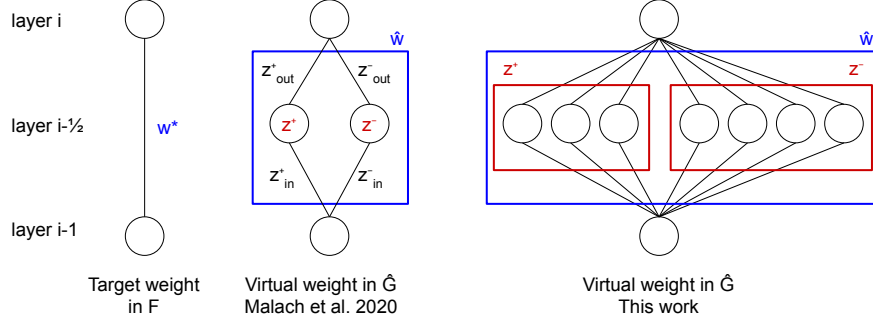

Figure 1: The target weight $w^*$ is simulated in the *pruned* network $\hat{G}$ by 2 intermediate neurons, requiring $1/\varepsilon^2$ sampled neurons (previous work) or by $2 \log 1/\varepsilon$ intermediate neurons due to a 'binary' decomposition of $w^*$, requiring only $O(\log 1/\varepsilon)$ sampled neurons (this work).

**Weight decomposition.** Our most important improvement is to build the weight $\hat{w}$ not with just two intermediate neurons, but with $O(\log {}^1\!/_\varepsilon)$ of them, so as to decompose the weight into pieces of different precisions, and recombine them with the sum in the neuron at layer $i+1$ (see Fig. 1), using a suitable binary mask vector $\boldsymbol{b}$ in the pruned network $\hat{G}$. Intuitively, the weight $\hat{w}$ is decomposed into its binary representation up to a precision of $k \approx \lceil \log_2 1/\varepsilon \rceil$ bits: $\sum_{s=1}^{k} b_s 2^{-s}$. Using a uniform distribution to obtain these weights $2^{-s}$ would not help, however. But, because the high precision bits are now all centered around 0, we can use a hyperbolic sampling distribution $p_w(|w|) \propto 1/w$ which has high density near 0. More precisely, but still a little simplified, for a weight $w^* \in [-1, 1]$ we approximate $w^*$ within $\approx 2^{-k}$ accuracy with the virtual weight $\hat{w}$ such that:

$$\hat{w}x = \sum_{s=1}^{k} b_s \left[ z_{\text{out},s}^+ \sigma(z_{\text{in},s}^+ x) + z_{\text{out},s}^- \sigma(z_{\text{in},s}^- x) \right] \approx \sum_{s=1}^{k} b_s \text{sgn}(w^*) 2^{-s} x \approx w^* x \qquad (1)$$

where $b_s \in \{0, 1\}$ is factored out since all connections have the same mask, and where $z_{\text{out},s}^+ z_{\text{in},s}^+ \approx \text{sgn}(w^*) 2^{-s} \approx z_{\text{out},s}^- z_{\text{in},s}^-$ and $z_{\text{out},s}^+ > 0$, $\text{sgn}(z_{\text{in},s}^+) = \text{sgn}(w^*)$, $z_{\text{out},s}^- < 0$ and $z_{\text{in},s}^- = -\text{sgn}(w^*)$. Note however that, because of the inexactness of the sampling process, we use a decomposition in base ${}^3\!/_2$ instead of base 2 (Lemma 9 in Section 7).

**Product weights.** Recall that $z_{\text{out},s}^+ \sigma(z_{\text{in},s}^+ x) = z_{\text{out},s}^+ \max\{0, z_{\text{in},s}^+ x\}$. For fixed signs of $z_{\text{out},s}^+$ and $z_{\text{in},s}^+$, this function can be equivalently calculated for all possible values of these two weights such that the product $z_{\text{out},s}^+ z_{\text{in},s}^+$ remains unchanged. Hence, forcing $z_{\text{out},s}^+$ and $z_{\text{in},s}^+$ to take 2 specific values is wasteful as one can take advantage of the cumulative probability mass of all their combinations. We thus make use of the induced product distribution, which avoids squaring the number of required samples. Define the distribution $p_{w \geq 0}$ for positive weights $w \in [\alpha, \beta]$ with $0 < \alpha < \beta$ and $p_w$, symmetric around 0, for $w \in [-\beta, -\alpha] \cup [\alpha, \beta]$:

$$p_{w \geq 0}(w) = \frac{1}{w \ln(\beta/\alpha)} \propto \frac{1}{w}, \quad \text{and} \quad p_w(w) = p_w(-w) = \tfrac{1}{2} p_{w \geq 0}(|w|) = \frac{1}{2|w| \ln(\beta/\alpha)}.$$

Then, instead of sampling uniformly until both $z_{\text{out},s}^+ \approx 1$ and $z_{\text{in},s}^+ \approx w^*$, we sample both from $p_w$ so that $z_{\text{out},s}^+ z_{\text{in},s}^+ \approx w^*$, taking advantage of the induced product distribution $p_{w \times} \approx \tfrac{1}{2} p_{w \geq 0}$ (Lemma C.11).

**Batch sampling.** Sampling sufficiently many intermediate neurons so that a subset of them are employed in approximating one target weight $w^*$ with high probably and then discarding (pruning) all other intermediate neurons is wasteful. Instead, we allow these samples to be 'recycled' to be used for other neurons in the same layer. This is done by partitioning the neurons in different buckets (categories) and ensuring that each bucket has enough neurons (Lemma C.12).

# 6 Theoretical Results

We now have all the elements to present our central theorem, which tells us how many intermediate neurons to sample to approximate all weights at a layer of the target network with high probability. Remark 4 below will then describe the result in terms of number of neurons per target weight.

**Theorem 3** (ReLU sampling bound). *For a given architecture $A(\ell, \mathbf{n}, \boldsymbol{\sigma})$ where $\sigma$ is the ReLU function, with weights in $[-w_{\max}, w_{\max}]$ and a given accuracy $\varepsilon$, the network $G$ constructed as above with weights sampled from $p_w$ with $[\alpha, \beta] = [\alpha'/q, \beta'/q], \alpha' = 2\varepsilon_w/9, \beta' = 2w_{\max}/3$, and $q = (\alpha'\beta')^{1/4}$, requires only to sample $M_i$ intermediate neurons for each layer $i$, where*

$$M_i = \left\lceil 16k'\left(n_i n_{i-1} + \ln\frac{2\ell k'}{\delta}\right)\right\rceil \quad \text{with} \quad k' = \log_{3/2}\frac{3w_{\max}}{\varepsilon_w} \quad \text{and}$$

$$\varepsilon_w = \varepsilon \bigg/ \left(e\,\ell\, n_{\max}^{3/2}\, F_{\max}(\mathcal{X}) \prod_{i=1}^{\ell} \max\{1, \|\hat{W}_i\|_2\}\right)$$

*($\varepsilon_w$ is in Lemma 1 with $\lambda = 1$ for ReLU), in order to ensure that for any target network $F$ with the given architecture $A(\ell, \mathbf{n}, \boldsymbol{\sigma})$, there exist binary masks $\boldsymbol{b}_{i,j} = (b_{i,j,1}, \ldots b_{i,j,n_{i-1}})$ of $G$ such that for the resulting subnetwork $\hat{G}$,*

$$\sup_{x \in \mathcal{X}} \|F(x) - \hat{G}(x)\|_2 \leq \varepsilon.$$

*Proof.* **Step 1. Sampling intermediate neurons to obtain product weights.** Consider a single target weight $w^*$. Recalling that $z_{\text{out},s}^+ > 0$ and $z_{\text{out},s}^- < 0$, we rewrite Eq. (1) as

$$\hat{w}x = \sum_{s=1}^{k} b_s z_{\text{out},s}^+ \sigma(z_{\text{in},s}^+ x) + \sum_{s=1}^{k} b_s z_{\text{out},s}^- \sigma(z_{\text{in},s}^- x)$$

$$= \sum_{s=1}^{k} b_s \sigma(\underbrace{z_{\text{out},s}^+ z_{\text{in},s}^+}_{\hat{w}^+} x) + \sum_{s=1}^{k} -b_s \sigma(-\underbrace{z_{\text{out},s}^- z_{\text{in},s}^-}_{\hat{w}^-} x)$$

The two virtual weight $\hat{w}^+$ and $\hat{w}^-$ are obtained separately. We need both $|w^* - \hat{w}^+| \leq \varepsilon_w/2$ and $|w^* - \hat{w}^-| \leq \varepsilon_w/2$ so that $|w^* - \hat{w}| \leq \varepsilon_w$.

Consider $\hat{w}^+$ (the case $\hat{w}^-$ is similar). We now sample $m$ intermediate neurons, fully connected to the previous and next layers, but only keeping the connection between the same input and output neurons as $w^*$ (the other weights are zeroed out by the mask $\boldsymbol{b}$). For a single sampled intermediate neuron $z$, all its weights, including $z_{\text{in}}^+$ and $z_{\text{out}}^+$, are sampled from $p_w$, thus the product $|z_{\text{out}}^+ z_{\text{in}}^+|$ is sampled from the induced product distribution $p_{w\times}$ and, a quarter of the time, $z_{\text{out}}^+$ and $z_{\text{in}}^+$ have the correct signs (recall we need $z_{\text{out}}^+ > 0$ and $\text{sgn}(z_{\text{in}}^+) = \text{sgn}(w^*)$). Define

$$p^+(z_{\text{out}}^+ z_{\text{in}}^+) = P(w = z_{\text{out}}^+ z_{\text{in}}^+ \ \wedge \ z_{\text{out}}^+ \sim p_w \ \wedge \ z_{\text{out}}^+ > 0$$
$$\wedge \ z_{\text{in}}^+ \sim p_w \ \wedge \ \text{sgn}(z_{\text{in}}^+) = \text{sgn}(w^*))$$

then with $p^+(z_{\text{out}}^+ z_{\text{in}}^+) \geq p_{w\times}(|z_{\text{out}}^+ z_{\text{in}}^+|)/4 \geq p_{w\geq 0}(|z_{\text{out}}^+ z_{\text{in}}^+|)/8$ where the last inequality follows from Lemma C.11 for $|z_{\text{out}}^+ z_{\text{in}}^+| \in [\alpha', \beta']$, $z_{\text{out}}^+ \in [\alpha, \beta]$ and $z_{\text{in}}^+ \in [\alpha, \beta]$, and similarly for $z_{\text{out}}^- z_{\text{in}}^-$ with $p^-(z_{\text{out}}^- z_{\text{in}}^-) \geq p_{w\geq 0}(|z_{\text{out}}^- z_{\text{in}}^-|)/8$.

Note that because $\text{sgn}(z_{\text{out}}^+) = -\text{sgn}(z_{\text{out}}^-)$ and $\text{sgn}(z_{\text{in}}^+) = -\text{sgn}(z_{\text{in}}^-)$, the samples for $\hat{w}^+$ and $\hat{w}^-$ are mutually exclusive which will save us a factor 2 later.

**Step 2. 'Binary' decomposition/recomposition.** Consider a target weight $w^* \in [-w_{\max}, w_{\max}]$. Noting that Corollary 10 equally applies for negative weights by first negating them, we obtain $\hat{w}^+$ and $\hat{w}^-$ by two separate applications of Corollary 10 where we substitute $P_\varepsilon \rightsquigarrow P_\varepsilon/8 = p_{w\geq 0}/8$, $\varepsilon \rightsquigarrow \varepsilon_w/2, \delta \rightsquigarrow \delta_w$. Substituting $P_\varepsilon$ with $p_{w\geq 0}/8$ in Eq. (2) shows that this leads to a factor 8 on $m$. Therefore, by sampling $m = 8\lceil k' \ln\frac{k'}{\delta_w}\rceil$ weights from $p_{w\times}$ in $[\alpha', \beta'] = [2\varepsilon_w/9, 2w_{\max}/3]$ with $k' = \log_{3/2}\frac{3w_{\max}}{\varepsilon_w}$ ensures that there exists a binary mask $\boldsymbol{b}$ of size at most $k'$ such that $|w^* - \hat{w}^+| \leq \varepsilon_w/2$ with probability at least $1 - \delta_w$. We proceed similarly for $w^-$. Note that Corollary 10 guarantees $|\hat{w}| \leq |w^*| \leq w_{\max}$, even though the large network $G$ may have individual weights larger than $w_{\max}$.

**Step 2'. Batch sampling.** Take $k := \lceil \log_{3/2} \frac{w_{\max}}{2\varepsilon_w} \rceil \leq k'$ to be the number of 'bits' required to decompose a weight with Corollary 10 (via Lemma 9). Sampling $m$ different intermediate neurons for each target weight and discarding $m - k$ samples is wasteful: Since there are $n_i n_{i-1}$ target weights at layer $i$, we would need $n_i n_{i-1} m$ intermediate neurons, when in fact most of the discarded neurons could be recycled for other target weights.

Instead, we sample all the weights of layer $i$ at the same time, requiring that we have at least $n_i n_{i-1}$ samples for each of the $k$ intervals of the 'binary' decompositions of $\hat{w}^+$ and $\hat{w}^-$. Then we use Lemma C.12 with $2k$ categories: The first $k$ categories are for the decomposition of $\hat{w}^+$ and the next $k$ ones are for $\hat{w}^-$. Note that these categories are indeed mutually exclusive as explained in Step 1. and, adapting Eq. (2), each has probability at least $\frac{1}{8} \int_{w=\gamma^{u+1}}^{\gamma^u} p_{w \geq 0}(w) dw \geq 1/(8 \log_{3/2}(3 w_{\max}/\varepsilon_w)) = 1/(8k')$ (for any $u$). Hence, using Lemma C.12 where we take $n \rightsquigarrow n_i n_{i-1}$ and $\delta \rightsquigarrow \delta_i$, we only need to sample $\lceil 16k'(n_i n_{i-1} + \ln \frac{2k}{\delta_i}) \rceil \leq \lceil 16k'(n_i n_{i-1} + \ln \frac{2k'}{\delta_i}) \rceil = M_i$ intermediate neurons to ensure that with probability at least $1 - \delta_i$ each $\hat{w}^+$ and $\hat{w}^-$ can be decomposed into $k$ product weights in each of the intervals of Lemma 9.

**Step 3. Network approximation.** Using a union bound, we need $\delta_i = \delta/\ell$ for the claim to hold simultaneously for all $\ell$ layers. Finally, when considering only the virtual weights $\hat{w}$ (constructed from $\hat{w}^+$ and $\hat{w}^-$), $\hat{G}$ and $F$ now have the same architecture, hence choosing $\varepsilon_w$ as in Lemma 1 ensures that with probability at least $1 - \delta$, $\sup_{x \in \mathcal{X}} \|F(x) - \hat{G}(x)\| \leq \varepsilon$. $\square$

**Remark 4.** Consider $n_i = n_{\max}$ for all $i$ and assume $\|W_i\|_2 \leq 1$, $w_{\max} = 1$ and $F_{\max}(\mathcal{X}) \leq 1$. Then $\varepsilon_w \geq \varepsilon/(e\ell n_{\max}^{3/2})$ and $k' \leq \log_{3/2}(3e\ell n_{\max}^{3/2}/\varepsilon)$. Then we can interpret Theorem 3 as follows: When sampling the weights of a ReLU architecture from the hyperbolic distribution, we only need to sample $M_i/n_{\max}^2 \leq 16k' + \ln(2\ell k'/\delta)/n_{\max}^2 = \tilde{O}(\log(\ell n_{\max}/\varepsilon))$ neurons per target weight (assuming $n_{\max}^2 > \log(\ell k'/\delta)$). Compare with the bound of Malach et al. [2020, Theorem A.6] which, under the further constraints that $w_{\max} \leq 1/\sqrt{n_{\max}}$ and $\max_{i \in [\ell]} \|W_i^*\|_0 \leq n_{\max}$ and with uniform sampling in $[-1, 1]$, needed to sample $M_i/n_{\max}^2 = \lceil 64\ell^2 n_{\max}^3 \log(2N/\delta)/\varepsilon^2 \rceil$ neurons per target weight.

**Example 5.** Under the same assumptions as Remark 4, for $n_{\max} = 100$, $\ell = 10$, $\varepsilon = 0.01$, $\delta = 0.01$, the bound above for Malach et al. [2020] gives $M_i/n_{\max}^2 \leq 2 \cdot 10^{15}$, while our bound in Theorem 3 gives $M_i/n_{\max}^2 \leq 630$. $\triangle$

**Example 6.** Under the same conditions as Example 5, if we remove the assumption that $\|W_i\|_2 \leq 1$, then Theorem 3 gives $M_i/n_{\max}^2 = \tilde{O}(\ell \log(n_{\max}/\varepsilon))$ and numerically $M_i/n_{\max}^2 \leq 2\,450$. $\triangle$

We can now state our final result.

**Corollary 7** (Weight count ratio). Under the same conditions as Theorem 3, Let $N^*$ be the number of weights in the fully connected architecture $A(\ell, \mathbf{n}, \boldsymbol{\sigma})$ and $N_G$ the number of weights of the large network $G$, then the weight count ratio is $N_G/N^* \leq 32 n_{\max} k' + \tilde{O}(\log(k'/\delta))$.

*Proof.* We have $N^* = \sum_{i=1}^{\ell} n_{i-1} n_i$, and the total number of weights in the network $G$ if layers are fully connected is at most $N_G = \sum_{i=1}^{\ell} (n_{i-1} + n_i) M_i$, where $M_i = 16k' n_{i-1} n_i + O(\log(k'/\delta))$. Hence the weight count ratio is $N_G/N^* \leq 32 n_{\max} k' + \tilde{O}(\log(k'/\delta))$. $\square$

**Remark 8.** Since in the pruned network $\hat{G}$ each target weight requires $k'$ neurons, the large network has at most a constant factor more neurons than the pruned network.

# 7  Technical lemma: Random weights

The following lemma shows that if $m$ weights are sampled from a hyperbolic distribution, we can construct a 'goldary' (as opposed to 'binary') representation of the weight with only $\tilde{O}(\ln \frac{1}{\varepsilon} \ln \frac{1}{\delta})$ samples. Because of the randomness of the process, we use a "base" $3/2$ instead of a base $2$ for logarithms, so that the different 'bits' have overlapping intervals. As the proof clarifies, the optimal base is $1/\gamma = \frac{1}{2}(\sqrt{5} + 1) \doteq 1.62$. The base $1/\gamma = 3/2$ is convenient. The number $\frac{1}{2}(\sqrt{5} + 1)$ is known as the 'golden ratio' in the mathematical literature, which explains the name we use.

**Lemma 9** (Golden-ratio decomposition). For any given $\varepsilon > 0$ and $1/\varphi \le \gamma < 1$, where $\varphi := \frac{1}{2}(\sqrt{5}+1)$ is the golden ratio, define the probability density $P_\varepsilon(v) := \frac{c'}{v}$ for $v \in [\varepsilon\gamma^2, \gamma]$ with normalization $c' := [\ln\frac{1}{\gamma\varepsilon}]^{-1}$. For any $\delta \in (0,1)$, if $m = \lceil k'\ln k'/\delta \rceil = \tilde{\Omega}(\ln\varepsilon \cdot \ln\delta)$ with $k' := \log_\gamma(\gamma\varepsilon)$, then with probability at least $1-\delta$ over the random sampling of $m$ 'weights' $v_s \sim P_\varepsilon$ for $s = 1,...,m$, the following holds: For every 'target weight' $w \in [0,1]$, there exists a mask $\boldsymbol{b} \in \{0,1\}^m$ with $|\boldsymbol{b}| \le k'$ such that $\hat{w} := b_1 v_1 + ... + b_m v_m$ is $\varepsilon$-close to $w$, indeed $w - \varepsilon \le \hat{w} \le w$.

*Proof.* Let $k = \lceil \log_\gamma \varepsilon \rceil \le 1 + \log_\gamma \varepsilon = k'$. First, consider a sequence $(v_i)_{i\in[k]}$ such each $v_i$ is in the interval $I_i := (\gamma^{i+1}, \gamma^i]$ for $i = 1,...,k$. We construct an approximating $\hat{w}$ for any weight $w_0 := w \in [0,1]$ by successively subtracting $v_i$ when possible. Formally

$$\text{for}(i = 1,...,k) \ \{\text{if } w_{i-1} \ge \gamma^i \text{ then } \{w_i := w_{i-1} - v_i; \ b_i = 1\} \text{ else } \{w_i := w_{i-1}; \ b_i = 0\}\}$$

By induction we can show that $0 \le w_i \le \gamma^i$. This holds for $w_0$. Assume $0 \le w_{i-1} \le \gamma^{i-1}$: If $w_{i-1} < \gamma^i$ then $w_i = w_{i-1} < \gamma^i$.

$$\text{If } w_{i-1} \ge \gamma^i \text{ then } w_i = w_{i-1} - v_i \le \gamma^{i-1} - \gamma^{i+1} = (\gamma^{-1} - \gamma)\gamma^i \le \gamma^i.$$

The last inequality is true for $\gamma \ge \frac{1}{2}(\sqrt{5}-1)$, which is satisfied due to the restriction $1/\varphi \le \gamma < 1$. Hence the error $0 \le w - \hat{w} = w_k \le \gamma^k \le \varepsilon \le \gamma^{k-1}$ for $k := \lceil \log_\gamma \varepsilon \rceil \ge 0$.

Now consider a random sequence $(v_i)_{i\in[m]}$ where we sample $v_s \overset{iid}{\sim} P$ over the interval $[\gamma^2\varepsilon, \gamma]$ for $s = 1,...,m > k$. In the event that there is at least one sample in each interval $I_i$, we can use the construction above with a subsequence $\tilde{v}$ of $v$ such that $\tilde{v}_i \in I_i$ and $\sum_{i\in[k]} b_i\tilde{v}_i = w_k$ as in the construction above. Next we lower bound the probability $p$ that each interval $I_i$ contains at least one sample. Let $E_i$ be the event "no sample is in $I_i$" and let $c = \min_{i\in[k]} P[v \in I_i]$. Then $P[E_i] = (1 - P[v \in I_i])^m \le (1-c)^m$, hence

$$p = 1 - P[E_1 \vee ... \vee E_k] \ge 1 - \sum_{i=1}^{k} P[E_i] \ge 1 - k(1-c)^m \ge 1 - k\exp(-cm)$$

and thus choosing $m \ge \lceil \frac{1}{c}\ln(k/\delta) \rceil$ ensures that $p \ge 1-\delta$. Finally,

$$c = \min_{i\in[k]} P[v \in I_i] = \min_i P[\gamma^{i+1} < v \le \gamma^i] = \min_i \int_{\gamma^{i+1}}^{\gamma^i} \frac{c'}{v}dv = c'\ln\frac{1}{\gamma} = 1/\log_\gamma(\gamma\varepsilon) = \frac{1}{k'} \quad (2)$$

and so we can take $m = \lceil k'\ln\frac{k'}{\delta} \rceil$. $\square$

**Corollary 10** (Golden-ratio decomposition for weights in $[0, w_{\max}]$). For any given $\varepsilon > 0$, define the probability density $P_\varepsilon(v) := \frac{c'}{v}$ for $v \in [\frac{4}{9}\varepsilon, \frac{2}{3}w_{\max}]$ with normalization $c' := 1/\ln\frac{3w_{\max}}{2\varepsilon}$. Let $k' := \log_{3/2}\frac{3w_{\max}}{2\varepsilon}$, For any $\delta \in (0,1)$, if $m = \lceil k'\ln\frac{k'}{\delta} \rceil = \tilde{\Omega}(\ln\frac{1}{\varepsilon} \cdot \ln\frac{1}{\delta})$, then with probability at least $1-\delta$ over the random sampling of $m$ 'weights' $v_s \sim P_\varepsilon$ ($s = 1,...,m$) the following holds: For every target 'weight' $w \in [0, w_{\max}]$, there exists a mask $\boldsymbol{b} \in \{0,1\}^m$ with $|\boldsymbol{b}| \le k'$ such that $\hat{w} := b_1 v_1 + ... + b_m v_m$ is $\varepsilon$-close to $w$, indeed $w - \varepsilon \le \hat{w} \le w$.

*Proof.* Follows from Lemma 9 with $\gamma = 2/3$ and a simple rescaling argument: First rescale $w' = w/w_{\max}$ and apply Lemma 9 with $w'$ and accuracy $\varepsilon/w_{\max}$. Then the constructed $\hat{w}'$ satisfies $w' - \varepsilon/w_{\max} \le \hat{w}' \le w'$ and multiplying by $w_{\max}$ gives the required accuracy. Also note that the density $P_\varepsilon(v) \propto 1/v$ is scale-invariant. $\square$

# 8 Related Work

In the version of this paper that was submitted for review, we conjectured with supporting experimental evidence that high precision could be obtained also with uniform sampling when taking advantage of sub-sums (see Appendix A). After the submission deadline, we have been made aware that Pensia et al. [2020] concurrently and independently submitted a paper that resolves this conjecture, by using a theorem of Lueker [1998]. Pensia et al. [2020] furthermore use a different grouping of the samples in each layer, leading to a refined bound with a logarithmic dependency on the number of *weights* per target weight and provide a matching lower bound (up to constant factors). Their results are heavily anchored in the assumptions that the max norms of the weights and of the inputs are bounded by 1, leaving open the question of what happens without these constraints—this could be dealt with by combining their results with our Lemma 1.

# 9 Conclusion

We have proven that large randomly initialized ReLU networks contain many more subnetworks than previously shown, which gives further weight to the idea that one important task of stochastic gradient descent (and learning in general) may be to effectively prune connections by driving their weights to 0, revealing the so-called winning tickets. One could even conjecture that the effect of pruning is to reach a vicinity of the global optimum, after which gradient descent can perform local quasi-convex optimization. Then the required precision $\varepsilon$ may not need to be very high.

Further questions include the impact of convolutional and batch norm layers, skip-connections and LSTMs on the number of required sampled neurons to maintain a good accuracy.

## Statement of broader impact

This work is theoretical, and in this regard we do not expect any direct societal or ethical consequences. It is our hope, however, that by studying the theoretical foundations of neural networks this will eventually help the research community make better and safer learning algorithms.

### Acknowledgements

The authors would like to thank Claire Orseau for her great help with time allocation, Tor Lattimore for the punctual help and insightful remarks, András György for initiating the Neural Net Readathon, and Ilja Kuzborskij for sharing helpful comments.

## Footnotes

[1]Note that even though their bounds are stated in the 1-norm, this is because they consider a single output—for multiple outputs their result holds in the 2-norm, which is what their proof uses.

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
