[Supplementary Material]

# A Sub-sums of Uniform Samples

Sampling uniformly requires many samples to obtain high precision anywhere in the region of interest (Fig. 2, Top). In this work we have taken advantage the summing function in neurons, combined with pruning, so as to be able to consider all $2^k$ sub-sums of $k$ samples (Fig. 2, Middle). However, we conjecture that a similar effect appears with sub-sums of uniform samples (Fig. 2, Bottom, but observe the large offset): For example, it suffices that 2 among $k$ samples $x_1$ and $x_2$ are within $\varepsilon$ of each other so that for all other samples $x_3$, $x_3 + x_1$ and $x_3 + x_2$ are within $\varepsilon$ of each other too.

# B Variations and Improvements

**Remark B.1.** Sampling from $P_\varepsilon$ is easy using the inverse CDF: To obtain a sample from $P_\varepsilon$ in $[\alpha, \beta]$, first draw a uniform sample $u \sim \mathcal{U}[0, 1]$ then return $\alpha(\beta/\alpha)^u$.

**Remark B.2.** The sampling procedure can be made independent of $\varepsilon$ by sampling from $P(v) = \ln \frac{1}{\gamma} / v (\ln v)^2$ for $0 \le v \le \gamma$ with $\gamma = \frac{2}{3}$. It is easy to see that $c$ in the proof becomes $(\ln \gamma / \ln(\varepsilon \gamma^2))^2$, leading to a slightly worse bound $m = \tilde{\Omega}((\ln \frac{1}{\varepsilon})^2 \cdot \ln \frac{1}{\delta})$. For $P(v) \propto [v \ln \frac{1}{v} (\ln(\ln \frac{3}{v}))^2]^{-1}$ we get the same bound $m = \tilde{\Omega}(\ln \frac{1}{\varepsilon} \cdot \ln \frac{1}{\delta})$ as in Lemma 9.

**Remark B.3.** Instead of using batch sampling and Lemma C.12, we can 'recycle' samples in a different way, which removes the leading factor 16 at the expense of a larger second order term. See Appendix D.

**Remark B.4.** What if $w_{\max}$ is not known? A simple trick is to take $w_{\max} = 2^{j+1}$ with $j \sim P(j) \propto 1/(j \ln^2(j+1))$. Then the total number of samples required to obtain $M_i^*$ samples in layer $i$ for the a priori unknown $w_{\max}^*$ is $O(M_i^* \log w_{\max}^* \log^2 \log w_{\max}^*)$.

**Remark B.5.** In practice, weights are often initialized uniformly in $[-O(\sqrt{1/n}), +O(\sqrt{1/n})]$, where $n$ is the layer width, potentially somehow averaged over two layers, i.e. weights are initially very small. All our initializations need *some* large weights but only very few ($O(\log n)$ outside this interval), most weights are very small too. We could even eliminate the large weights and limit our sampling procedure to weights in this interval, but sample $O(\sqrt{n})$ times more weights to reconstruct large weights.

**Remark B.6.** Extreme case: Pruning 'Boolean' networks. The difficulty of pruning can be easily seen for 'Boolean' networks with a Heaviside transition function and binary inputs, and Boolean weights everywhere. Then, the network $G$ should need only twice as many weights as the target network, but can still represent exponentially many functions. It is then clear that not only "Pruning really is all you need," but also that "Pruning is as hard as learning."

# C Technical Results

**Proof of Lemma 1.**

Figure 2: y-axis: Difference between two consecutive points on the x-axis. Top: 1000 uniform samples in [0, 1], x-axis is sample value (previous work); Middle: 15 hyperbolic samples in [0, 1], each point is one of $2^{15}$ possible sub-sums, x-axis is the sub-sums (this work); Bottom: Like Middle, but with uniform samples in [0, 1] (future work?).

*Proof.* For all $x \in \mathcal{X}$, for a layer $i$:

$$\|F_i(x) - \hat{G}_i(x)\|_2 \stackrel{(a)}{=} \|f_i(F_{i-1}(x)) - \hat{g}_i(\hat{G}_{i-1}(x))\|_2$$

$$= \left[ \sum_{j \in [n_i]} (\sigma_i(W_{i,j}^* F_{i-1}(x)) - \sigma_i(\hat{W}_{i,j} \hat{G}_{i-1}(x)))^2 \right]^{1/2}$$

$$\stackrel{(b)}{\leq} \lambda_i \|W_i^* F_{i-1}(x) - \hat{W}_i \hat{G}_{i-1}(x)\|_2$$

$$\stackrel{(c)}{\leq} \lambda_i \|\hat{W}_i(F_{i-1}(x) - \hat{G}_{i-1}(x))\|_2 + \lambda_i \|(W_i^* - \hat{W}_i)F_{i-1}(x)\|_2$$

$$\stackrel{(d)}{\leq} \lambda_i \|\hat{W}_i\|_2 \|F_{i-1}(x) - \hat{G}_{i-1}(x)\|_2 + \lambda_i \|W_i^* - \hat{W}_i\|_2 \|F_{i-1}(x)\|_2$$

$$\stackrel{(e)}{\leq} \lambda_i \|\hat{W}_i\|_2 \|F_{i-1}(x) - \hat{G}_{i-1}(x)\|_2 + \lambda_{\max} \sqrt{n_i n_{i-1}} \varepsilon_w w_{\max} \sqrt{n_i} F_{\max}(x)$$

$$\leq \lambda_i \|\hat{W}_i\|_2 \|F_{i-1}(x) - \hat{G}_{i-1}(x)\|_2 + \varepsilon_w \lambda_{\max} n_{\max}^{3/2} F_{\max}(x)$$

$$\stackrel{(f)}{\leq} \varepsilon_w e i \lambda_{\max} n_{\max}^{3/2} F_{\max}(x) \prod_{u=1}^{i} \max\{1, \lambda_u \|\hat{W}_u\|_2\}$$

where (a) follows from the definition of $F_i$ and $\hat{G}_i$, (b) follows from $|\sigma_i(x) - \sigma_i(y)| \leq \lambda_i |x - y|$ by the definition of $\lambda_i$, (c) follows from the Minkowski inequality, (d) follows from $\|Mv\|_2 \leq \|M\|_2 \|v\|_2$ applied to both terms, (e) is by assumption that $|w^* - \hat{w}| \leq \varepsilon_w$ and $\|M\|_2 \leq c\sqrt{ab}$ for any $M \in [-c, c]^{a \times b}$, and finally (f) follows from Corollary C.10, using $\|F_0(x) - \hat{G}_0(x)\|_2 = 0$. Therefore

$$\|F(x) - \hat{G}(x)\|_2 = \|F_\ell(x) - \hat{G}_\ell(x)\|_2 \leq \varepsilon_w e \ell \lambda_{\max} n_{\max}^{3/2} F_{\max}(x) \prod_{i=1}^{\ell} \max\left\{1, \lambda_i \|\hat{W}_i\|_2\right\}$$

and taking $\varepsilon_w$ as in the theorem statement proves the result. $\square$

**Lemma C.1** (Bound on positive sequences). *Assuming $x_0 \geq 0$, and if, for all $t = 0, 1, \ldots, x_t \leq a_t x_{t-1} + b_t$ with $a_t \geq 0$ and $b_t \geq 0$, then*

$$\forall \tau \text{ s.t. } |\{a_t < 1 + 1/\tau\}_{t \in [T]}| \leq \tau \quad \text{we have} \quad x_T \leq e(x_0 + c) \prod_{\substack{t \in [T] \\ a_t \geq 1 + 1/\tau}}^{T} a_t$$

*with $c = \min\left\{\tau \max_t b_t, \ \max_t \dfrac{b_t}{a_t - 1}\right\}$.*

*Proof.* First, observe that $\tau \geq 0$, with $\tau = 0$ iff $T = 0$. Let $\tilde{a}_t = \max\{a_t, 1 + 1/\tau\}$. Then

$$c = \min\left\{\tau \max_t b_t, \ \max_t \frac{b_t}{a_t - 1}\right\} = \max_t \frac{b_t}{\max\{1/\tau, a_t - 1\}} = \max_t \frac{b_t}{\tilde{a}_t - 1}.$$

Define $y_t = \tilde{a}_t y_{t-1} + (\tilde{a}_t - 1)c$ and $y_0 = x_0$. Then we have $y_t + c = \tilde{a}_t(y_{t-1} + c)$ and so by recurrence $y_T + c = (y_0 + c) \prod_{t=1}^{T} a_t$ and thus $y_T \leq (y_0 + c) \prod_{t=1}^{T} \tilde{a}_t$. Now, observe that $y_t \geq \tilde{a}_t y_{t-1} + b_t \geq 0$ and so by recurrence with base case $y_0 = x_0$, $x_T \leq y_T$.

Furthermore

$$\prod_{t \in [T]} \tilde{a}_t \leq \prod_{t:a_t < 1 + 1/\tau} (1 + 1/\tau) \prod_{t:a_t \geq 1 + 1/\tau} a_t \leq (1 + 1/\tau)^\tau \prod_{t:a_t \geq 1 + 1/\tau} a_t,$$

noting that $(1 + 1/\tau)^\tau \leq e$. $\square$

**Remark C.2.** The factor $e$ should be 1 if $\min_t a_t \geq 1 + 1/\tau$.

**Remark C.3.** $\tau = T$ is always feasible.

**Remark C.4.** If $\min_t a_t \geq 2$, then $\tau = 1$ is feasible.

**Corollary C.5** (Feasible $\tau$ for Lemma C.1). *In the context of Lemma C.1, for all $x > 1$ taking $\tau = \max\{1/(x-1), |\{a_t < x\}_{t \in [T]}|\}$ is feasible.*

*Proof.* Take $y = 1/(x-1)$, so $x = 1 + 1/y$ and $\tau = \max\{y, |\{a_t < 1 + 1/y\}_t|\}$. Thus $|\{a_t < 1 + 1/\tau\}_t| \leq |\{a_t < 1 + 1/y\}_t| \leq \tau$ as required. $\square$

**Remark C.6.** If $1 < \min_t a_t \leq x$ then taking $\tau = 1/(x-1)$ is feasible.

**Remark C.7.** Taking $\tau = \max\{1, |\{a_t < 2\}_t|\}$ is feasible.

**Remark C.8.** For $\varphi = (1 + \sqrt{5})/2 \leq 1.62$, taking $\tau = \max\{\varphi, |\{a_t < \varphi\}_t|\}$ is feasible.

**Remark C.9.** If $\min_t a_t > 1$ then $\tau = 1/(\min_t a_t - 1)$ is feasible (but useful only if $\min_t a_t \geq 1 + 1/T$).

**Corollary C.10** (Simpler bound on positive sequences). *Assuming $x_0 = 0$, and if, for all $t = 0, 1, \ldots,$ $x_t \leq a_t x_{t-1} + b_t$ with $a_t \geq 0$ and $b_t \geq 0$, then*

$$x_T \leq eT \max_t b_t \prod_{t \in [T]} \max\{1, a_t\}.$$

*Proof.* Follows from Lemma C.1 with $\tau = T$ which is always feasible and observing that $\prod_{t:a_t \geq 1+1/T} a_t \leq \prod_t \max\{1, a_t\}$, and that $c \leq \tau \max_t b_t = T \max_t b_t$. $\square$

**Lemma C.11** (Product of weights). *Let probability densities $P_v(v) := c/v$ for $v \in [a, b]$ and $0 < a < b$ with normalization $c := 1/\ln\frac{b}{a}$. Let weight $w := v \cdot v'$ with $v$ and $v'$ both sampled from $P_v$. Then $P_w(w) \geq c/2w$ for $w \in [a', b']$, where $P_w$ is the probability density of $w$, and $a' := a\sqrt{ab}$ and $b' := b\sqrt{ab}$.*

Note that $w$ may be outside of $[a', b']$, but at least half of the time is inside $[a', b']$. The lemma implies that the bound in Lemma 9 also applies to the product of two weights, only getting a factor of 2 worse. Note that the scaling ranges $\frac{b}{a} = \frac{b'}{a'}$ are the same.

*Proof.* $\ln v$ is uniformly distributed in $[\ln a, \ln b]$: indeed, taking $y = \ln v$, we have $P_y(y) = P_v(v)/\frac{dy}{dv} = c$. Let us scale and shift this to $t := c(2 \ln v - \ln ab) \in [-1, +1]$ and similarly $t' := c(2 \ln v' - \ln ab) \in [-1, +1]$. Then $P_t(t) = P_v(v)/\frac{dt}{dv} = \frac{c}{v}/\frac{2c}{v} = \frac{1}{2}$ for $t \in [-1, +1]$, and same for $t'$. Let $u := t + t' \in [-2, +2]$. The sum of two uniformly distributed random variables is triangularly distributed: $P_u(u) = \frac{1}{2}(1 - \frac{1}{2}|u|)$. Using $w = v \cdot v'$ we can write $u = t + t' = 2c(\ln w - \ln(ab))$. Then $P_w(w) = P_u(u)\frac{du}{dw} = \frac{1}{2}(1 - \frac{1}{2}|u|)\frac{2c}{w}$. For $|u| \leq 1$ this is $\geq c/2w$. Finally $|u| \leq 1$ iff $|\ln w - \ln(ab)| \leq 1/2c$ iff $\ln w \gtrless \ln ab \mp \frac{1}{2}\ln\frac{b}{a}$ iff $w \in [a', b']$. $\square$

**Lemma C.12** (Filling $k$ categories each with at least $n$ samples). *Let $P_c$ be a categorical distribution of at least $k \in \mathbb{N}$ (mutually exclusive) categories $\{1, 2, \ldots k, \ldots\}$ such that the first $k$ categories have probability at least $c$ and at most $1/2$, that is, if $X \sim P_c$, then $c \leq P_c(X = j) \leq 1/2$ for all $j \in [k]$. Let $(X_i)_{i \in [M]}$ be a sequence of $M$ random variables sampled i.i.d. from $P_c$. For all $\delta \in (0, 1)$, for all $n \in \mathbb{N}$, if*

$$M = \left\lceil \frac{2}{c}\left(n + \ln\frac{k}{\delta}\right) \right\rceil$$

*then with probability at least $1 - \delta$ each category $j \in [k]$ contains at least $n$ samples, i.e., $|\{X_i = j\}_{i \in [M]}| \geq n$.*

*Proof.* Let $c_j \geq c$ be the probability of category $j \in [k]$. Using the Chernoff-Hoeffding theorem on the Bernoulli random variable $[\![X_i = j]\!]$, —where $[\![test]\!]$ is the indicator function and equals 1 if $test$ is true, 0 otherwise— with $Mc_j - x = n \geq 0$, that is, $x = Mc_j - n$, for each category $j \in [k]$ we

have

$$P\left(\sum_{i=1}^{M}(1-[\![X_i=j]\!]) > M(1-c_j)+x\right) \le \exp\left(-\frac{x^2}{2Mc_j(1-c_j)}\right)$$

$$P\left(\sum_{i=1}^{M}[\![X_i=j]\!] < Mc-x\right) \le \exp\left(-\frac{x^2}{2Mc_j(1-c_j)}\right)$$

$$P\left(\sum_{i=1}^{M}[\![X_i=j]\!] < n\right) \le \exp\left(-(Mc_j/2-n)\right)$$

and the condition $(1-c_j) \ge \mathrm{1/2}$ is satisfied. Name $E_j$ the event "the category $j \in [k]$ contains fewer than $n$ samples," then $P(E_j) \le \exp(-(Mc/2-n))$. Then, using a union bound, the probability that any of the $k$ categories contain fewer than $n$ samples is at most

$$P(E_1 \vee E_2 \vee \ldots E_k) \le \sum_{j=1}^{k} P(E_j) \le k\exp\left(-(Mc/2-n)\right)$$

and since $M \ge \frac{2}{c}\left(n+\ln\frac{k}{\delta}\right)$

$$P(E_1 \vee E_2 \vee \ldots E_k) \le \delta,$$
$$1 - P(E_1 \vee E_2 \vee \ldots E_k) \ge 1-\delta,$$

which proves the claim. $\qquad\square$

## D   Sample recycling

**Theorem D.1** (ReLU sampling bound #2)**.**  Theorem 3 holds simultaneously also with

$$M_i = \left\lceil 2k'\left(n_i n_{i-1} + 4\max\{n_i, n_{i-1}\}\ln\frac{2k'N^*}{\delta}\right)\right\rceil$$

and all other quantities are unchanged.

*Proof.* **Step 1 and 2.** Same as for Theorem 3.

**Step 2'. Sample recycling.** Let

$$m = \left\lceil 8k'\ln\frac{k'}{\delta_w}\right\rceil, \quad k' = \log_{3/2}\frac{3w_{\max}}{\varepsilon_w}, \quad k = \left\lceil\log_{3/2}\frac{2w_{\max}}{\varepsilon_w}\right\rceil,$$

where $m$ is the number of neurons that need to be sampled according to Corollary 10 to $\varepsilon_w$-approximate one target weight with probability at least $1-\delta_w$, and $k$ is an upper bound on the number of unit bits of the corresponding mask. One neuron with some pruned weights cannot be shared to approximate two target weights at the same time, which means we need at least $2kn_i n_{i-1}$ neurons ($k$ for each $\hat{w}^+$, and $k$ for each $\hat{w}^-$). For a specific target weight $w^*$, out of $m \ge 2k$ sampled neurons, only at most $2k$ of them are actually used to approximate the target weight; all others are 'discarded'. But discarding them is wasteful, because only the product weight on the same input/output as the target weight has been filtered by Corollary 10 (via Lemma 9); all other product weights are still independent samples since their values have not been queried by any process. Each intermediate neuron is connected in input and output with $n_i + n_{i-1}$ weights, but it contains exactly only $\min\{n_i, n_{i-1}\}$ independent *product* weight samples, since each input weight and each output weight can be used at most as one independent product weight sample. Algorithm 1 shows that we can use all of them and, following the algorithm's notation and the assumption that $n_i \ge n_{i-1}$, that for each $j$ we only need $m + 2(k-1)n_{i-1}$ sampled neurons, that is, only $n_i m + 2(k-1)n_i n_{i-1}$ for the whole layer. To also cover the case $n_{i-1} > n_i$, we need to sample $\max\{n_i, n_{i-1}\}m + 2(k-1)n_i n_{i-1}$ neurons to ensure that every target weight of layer $i$ can be decomposed into $2k$ product weights, each based on $m$ independent product weight samples.

**Step 3. Network approximation.** For the guarantee to hold simultaneously over all $\hat{w}^+$ and $\hat{w}^-$, using a union bound we can take $\delta_w = \delta/(2N^*)$. Finally the claim follows from Lemma 1 and noting that $k \le k'$. $\qquad\square$

**Remark D.2.** Observe that even though the factor in front of $m$ is larger than for Theorem 3, (also $\ell \rightsquigarrow N^*$ in the log) we gain a constant factor 8 in front of the leading term $n_i n_{i-1} k$.

**Example D.3.** Under the same conditions as Example 5, Theorem D.1 gives $M_i/n_{\max}^2 \leq 144$, and under the same conditions as Example 6 we have $M_i/n_{\max}^2 \leq 574$. △

Therefore, since both Theorem 3 and Theorem D.1 hold simultaneously, we can take:

$$M_i = \min\left\{ \left\lceil 16k' \left( n_i n_{i-1} + \ln \frac{2k'\ell}{\delta} \right) \right\rceil, \right.$$
$$\left. \left\lceil 2k' \left( n_i n_{i-1} + 4\max\{n_i, n_{i-1}\} \ln \frac{2k'N^*}{\delta} \right) \right\rceil \right\}$$

to ensure that, with probability at least $1 - \delta$,

$$\sup_{x \in \mathcal{X}} \|F(x) - \hat{G}(x)\|_2 \leq \varepsilon.$$

---

**Algorithm 1** Recycling samples. We assume that $n_i \geq n_{i-1}$, otherwise the loops and the increments need to be exchanged.

---

```
1  # Sample recycling at layer i.
2  for j= 1 to nᵢ: # Assumes nᵢ ≥ nᵢ₋₁
3     # Discard old samples and generate fresh ones.
4     M = sample m fully-connected intermediate neurons
5     for d = 1 to nᵢ₋₁:
6       # These indices ensure that
7       # * all target weights are approximated,
8       # * no input weight and no output weight is used for more
9       #   than one target weight.
10      idx_in = d
11      idx_out = (d+j) % nᵢ
12      w* = W*[idx_in, idx_out]
13      # 'Call' to the golden-ratio decomposition (Corollary 10)
14      # using the provided samples M.
15      # It returns the set K ⊆ M of sampled neurons used to decompose w*.
16      # Only uses weights at the indices idx_in and idx_out of the neurons
               in M.
17      # The indexes above ensure that no weight in M already has idx_in and
18      # idx_out zeroed out.
19      K+ = GRD+(M, idx_in, idx_out, w*) # Corollary 10 for ŵ⁺
20      K- = GRD-(M, idx_in, idx_out, w*) # Corollary 10 for ŵ⁻
21      # These samples cannot be reused for other neurons, put them aside.
22      M = M \ (K+ ∪ K-)
23
24      # Zero-out the input and output weights that the GRD has filtered,
25      # as they are not independent samples anymore and cannot be reused.
26      for n in M:
27        n.ins[idx_in] = 0
28        n.outs[idx_out] = 0
29
30      # Fill up M to have m intermediate neurons.
31      M_new = sample |K+|+|K-| new independent neurons   # |K+|+|K-| ≤ 2k
32      M = M ∪ M_new # such that |M| = m
```

---

# E  List of Notation

| Symbol | Explanation |
|---|---|
| $\mathbb{N}$ | natural numbers $\{1, 2, \ldots\}$ |
| $\ell \in \mathbb{N}$ | number of network layers |
| $\mathbf{n} \in \mathbb{N}^{\ell}$ | vector of the number of neurons |
| $n_{\max} \in \mathbb{N}$ | maximum number of neurons per layer |
| $i \in [\ell]$ | layer index |
| $j \in [n_i]$ | index of $j$th neuron in layer $i$ |
| $F$ | a target network |
| $G$ | the large network to be pruned |
| $\hat{G}$ | the network $G$ after pruning |
| $F_{\max}(\mathcal{X})$ | maximum absolute activation of any non-final neuron on all inputs of interest in $F$ |
| $w \in [-w_{\max}, w_{\max}]$ | some weight |
| $w_{\max} \in \mathbb{R}^+$ | max norm of the weights |
| $w^* \in [-w_{\max}, w_{\max}]$ | a weight of the target network $F$ |
| $W^*$ | weights of the target network |
| $z_{\text{out}}^+, z_{\text{in}}^+, z_{\text{out}}^-, z_{\text{in}}^-$ | actual individual weights of the network $G$ |
| $\hat{w}^+, \hat{w}^-, \hat{w}$ | virtual individual weights of the network $\hat{G}$ |
| $\varepsilon > 0$ | output accuracy |
| $1 - \delta \in [0, 1]$ | high probability |
| $\sigma : \mathbb{R} \to \mathbb{R}$ | activation function |
| $\boldsymbol{\sigma}$ | vector of $\ell$ activation functions |
| $\lambda_i$ | Lipschitz factor of $\sigma_i$ |
| $k \in \mathbb{N}$ | number of 'bits' to represent a weight |
| $m \in \mathbb{N}$ | number of neurons sampled per target weight |
| $M \in \mathbb{N}$ | number of neurons sampled per intermediate layer |
| $x \in [-x_{\max}, x_{\max}]^{n_0}$ | network input |
| $x_{\max} \in \mathbb{R}^+$ | max norm of the inputs |
| $P$ | probability |
| $A(\ell, \mathbf{n}, \boldsymbol{\sigma})$ | architecture of a network |
| $\boldsymbol{v}$ | vector |
| $\boldsymbol{b}$ | binary mask vector |
| $b$ | binary mask |
| $F_i$ | output of layer $i$ of the target network given network inputs |
| $G$ | the big network $i$ |
| $\hat{G}_i$ | subnetwork of the big network, approximating $F$ |
| $f_i$ | layer functions of target network given layer inputs |
| $\hat{g}_i$ | same as $f_i$ for $\hat{G}$ |
| $P_{\varepsilon}$ | $1/v$ distribution |
| $p_{w \geq 0}$ | $1/v$ distribution |
| $p_w$ | $\pm 1/v$ distribution |
| $p_{w \times}$ | $1/v$ product distribution |
| $p^+$ | product distribution of $z_{\text{out}}^+$ and $z_{\text{in}}^+$ |
| $p^-$ | product distribution of $z_{\text{out}}^-$ and $z_{\text{in}}^-$ |