[Reviews · NeurIPS 2020]

Review 1

Summary and Contributions: The paper shows that for some a target ReLU network F and a larger (overparametrized) network G, there exists a subnetwork of G of size greater than that of F by a factor logarithmic in all parameters of G except depth (possibly linear) that, without training, is a predictor that is with high probability close to F for all inputs. The work proceeds by constructing a bitwise (base 3/2) representation of weights in G to the desired accuracy; and exploiting the ReLU representation identity b*a*x = b*ReLU(a*x) – b*ReLU(-a*x) to sample from a product distribution for b*a. A covering argument for the support of a categorical distribution is used to show that the sampling is efficient, so that one pass of sampling can be done for all neurons in a given layer, circumventing a per-neuron cost in the similar construction of previous work. Finally, a union bound of error terms over all layers gives the result.

Strengths: The bound presented is an improvement on previous work, with assumptions that are no more stringent (notably boundedness assumptions on weights). The analogous construction of an intermediate layer of neurons for sampling between all true layers of G facilitates comparison and suggests that this is a useful approach to the problem. All of the techniques that enable this improvement are creative and quite clean. The bitwise representation of weights and product distribution for auxiliary weights both appear to be novel perspectives that may find use elsewhere.

Weaknesses: Some of the techniques are unusual, and the paper would benefit from more of a roadmap stating what will be presented where, and indications as to the actual content of various results if they are mentioned before presented (as is often the case, e.g., Lemma 9, Lemma 29). Equation numbering would also help: e.g., Equation between-lines-225-and-226 is quite difficult to parse.

Correctness: At this level of review, the paper generally appears correct. The use of base 2 for purposes of illustration (subsection “Weight Decomposition” in Section 5) does not actually work following the proof of Lemma 9 (cf. lines 294-295); it would be better to use base 3/2 from the start and state that the choice is explained via the proof of Lemma 9.

Clarity: The paper is generally well written, though organization could be improved, as noted above. Results stated in the supplementary material could perhaps also be labeled differently (e.g., Lemma A.1). As it is currently written, the reader sometimes encounters a reference to a semi-magical result and has to hunt for it in both the rest of the text and the appendix.

Relation to Prior Work: Yes.

Reproducibility: Yes

Additional Feedback:


Review 2

Summary and Contributions: This paper addresses the strong lottery ticket hypothesis -- the conjecture that every randomly initialized neural network can be pruned (i.e. have some of its weights zeroed out) to achieve good performance. The paper improves upon the state of the art in this area (of Malach et al. 2020) by showing that to approximate a target neural network, a randomly initialized neural network that is larger by a logarithmic factor contains such a pruning. While the paper reuses the central idea from Malach et al. of approximating each weight individually, the details of their approach differs substantially, including using a hyperbolic distribution over the random network weights to approximate the binary representation of the target weights, as opposed to uniform sampling as in Malach et al.

Strengths: The paper improves upon the results of Malach et al., reducing the blowup of the network size from a polynomial to a logarithmic factor.

Weaknesses: 1. The approach of the paper is significantly more complicated than that of Malach et al. 2. The hyperbolic sampling distribution does not reflect what is done in practice.

Correctness: I have not thoroughly checked the proofs, but the general proof idea seems sound.

Clarity: The paper is clear and well-written.

Relation to Prior Work: This paper appropriately frames itself in terms of previous work.

Reproducibility: Yes

Additional Feedback:


Review 3

Summary and Contributions: The paper presents an improved verion of the proof of the strong lottery ticket hypothesis presented by Malach et al. The improvement removes assumptions on the norms of the weights and inputs, and it reduces the requirements on the over-parameterisation of the width of the network from a polynomial factor to a logarithmic factor. The main construction is the same as in Malach et al. The main idea in the paper is that, instead of considering the event that 2 intermediate neurons immitate one weight of the target network, one considers the event that the combination of O(1/epsilon) intermediate neurons immitate one target weight. This combination is motivated by the idea of binary representations. To this end, two further results are needed and presented: When considering the event just described, there is no need to "discard" these intermediate neurons - instead the events are conidered together (which is solved via a partitioning of the neurons and showing that every partiton contains enough neurons). Another helpful observation is that events regarding products of weights can be considered (instead of analysing the weights (factors) separately).

Strengths: - The idea build the weights not just with two intermediate neurons as in the orignial work but with O(1/eps) is nice. - The motivation for construction (binary representation) is straightforward - yet the analysis is non-trivial. - Moreover, the paper improves the previously known bound on the overparameterisation significantly.

Weaknesses: - The main building blocks of introducing intermediate layers to mimic the weight of the target network and to prune weights, from Malach et al, are reused. No fundamentally new construction (except the different weight distribution) is presented and so it is a bit incremental. - It's still an open problem whether we can do better (lower bound). (https://arxiv.org/pdf/2006.07990.pdf (which appeared after submission deadline) seems to prove a lower bound on the required width for a 2-layer network). - Still limited implications: we know they exist but how to find good ones

Correctness: The (mostly high-level) proof in the main paper look good to me. Looked over some but not all the other proofs in the appendix.

Clarity: The paper is generally well written and easy to follow - with a main section that just describes the main ideas and overview on the construction; main result and technical lemmata are clearly described in separate section. (The idea of discarding the samples (which motivates the batch sampling) is a bit irritating imho, since all the weights are choosen iid.)

Relation to Prior Work: The paper gives a good overview on prior work and explains the improvements presented in the papers.

Reproducibility: Yes

Additional Feedback: Regarding the two paragraphs "Although our work assumes that the distribution of the weights is hyperbolic, we conjecture (with supporting empirical evidence in Appendix A) that a similar effect could be achieved with uniform samples. Combined with some of our proof ideas (batch sampling and product weights), it may be possible to reach the same type of bounds." (ll. 322): https://arxiv.org/pdf/2006.07990.pdf (which appeared after submission deadline) seems to do that. They propose the same idea of using multiple neurons, but make use of the "idea" of a (probabilistic) subset sum problem instead of binary representation (https://www.ics.uci.edu/~lueker/papers/exppart.pdf). (It looks a bit more elegant to me and more straightforward). However, they have to make assumptions on the norm of the weights and it doesn't look like one could get rid of this. Would be great if you could include comments in the final version. Post Author Feedback and Reviews: I've read the feedback and the other reviews and it remains an accept (regarding the "discarding" - just seemed natural to "overcomplicate" things right away instead of the naive approach when reading it (whereby I do *not* mean that I definitely could've come up with that stuff right away on my own))


Review 4

Summary and Contributions: This paper extends the insights on how to obtain the “winning ticket” from large neural network under assumption that the distribution of the weights is hyperbolic. The previous work is by Malach et al. [2020] showing that “pruning is all you need”: under some assumptions and uniform weight distribution, a large network that contains the winning ticket (small network having the same a or better accuracy than the large one) can be of size at most polynomial in the numbers of layers and neurons in the winning ticket. In contrast, this paper shows that under the hyperbolic weight distribution, the size of the large network is at most polylog of the number of layers and neurons “per target weight”. This is significantly lower than what shown by Malach et al. [2020]. The techniques are based on the construction of Malach et al. [2020] (mentioned in the paper) with new ideas on decomposition of weights, product weights, and sampling.

Strengths: This paper clearly defines the notation and gives the results of the previous wok and techniques it builds its results upon. Its new results also strengthen the previous work significantly, albeit with a limitation on the weight distribution of the neural network. However, it gives evidence that the results may hold for the uniform weight distribution. I believe this is a significant result that may deepen our understanding of the Lottery Ticket Hypothesis.

Weaknesses: It depends significantly on the construction of Malach et al. [2020] although the new decomposition technique is quite essential to obtain the logarithmic factor. I find that the exposition of the results is not clear. For example, at Line 66-68, the complexity is described per "target weight", but the results presented in Theorem 3 are not. I find it difficult to digest the results (Remark 4 should be mentioned earlier).

Correctness: The claims seem correct although I cannot follow all the proofs.

Clarity: The paper is clearly written, and the notation is sufficient to read and understand the results of the paper.

Relation to Prior Work: The paper gives enough background of the work and the previous results it builds upon.

Reproducibility: Yes

Additional Feedback:

[Author Response · NeurIPS 2020]

We thank the reviewers for their thoughtful observations and comments. Their input will be used to improve our paper. Overall, we are encouraged by the reviewers' reaction to our work. We don't see the need to go over the high level summary of our contributions at this time, given that all reviewers seem to be clear on this point. Next we address some outstanding comments, hoping that our responses clear any remaining concerns.

**Response to Reviewer 1**

*Regarding a roadmap, equation numbering, result labels*

These are all good ideas, thank you.

*At this level of review, the paper generally appears correct. The use of base 2 for purposes of illustration (subsection "Weight Decomposition" in Section 5) does not actually work following the proof of Lemma 9 (cf. lines 294-295); it would be better to use base 3/2 from the start and state that the choice is explained via the proof of Lemma 9.*

We chose base 2 because it makes it clear that the number can be efficiently decomposed. We expect more readers to be confused by another base. We can explain right after the binary decomposition that a base 3/2 is needed instead later. If the reviewer feels strongly about this, please let us know in the "post-rebuttal" so that we may reconsider.

**Response to Reviewer 2**

*(1) The approach of the paper is significantly more complicated than that of Malach et al.*
*(2) The hyperbolic sampling distribution does not reflect what is done in practice.*

Indeed our analysis is more complicated than that of Malach et al., but this is quite often the case for improvements on first simple theoretical results. We believe this is certainly worth the price. Regarding hyperbolic sampling, we agree that uniform sampling is more traditional and widespread, however our theory also suggests that hyperbolic sampling may be worth investigating in practice.

**Response to Reviewer 3**

*Regarding https://arxiv.org/pdf/2006.07990.pdf and https://www.ics.uci.edu/ lueker/papers/exppart.pdf:*

Thank you for these references. As you note, Pensia et al. came out after the submission deadline and we were indeed not aware of Lueker (1998). Both are quite nice pieces of work. Lueker's result is astounding and can indeed be used for uniform sampling, with the caveat that you note. We will certainly include discussions on these two papers in the final version.

*(The idea of discarding the samples (which motivates the batch sampling) is a bit irritating imho, since all the weights are choosen iid.)*

We are not entirely sure what precisely is irritating, but we can at least say early on that discarding samples is a naive approach if this helps.

**Response to Reviewer 4**

*I find that the exposition of the results is not clear. For example, at Line 66-68, the complexity is described per "target weight", but the results presented in Theorem 3 are not. I find it difficult to digest the results (Remark 4 should be mentioned earlier).*

Thank you for spotting some reading hurdles, which we will strive to fix. We will at the very least add a forward reference to Remark 4 in the revision.

[Meta-Review · NeurIPS 2020]

The paper shows that for some a target ReLU network F and a larger (overparametrized) network G, there exists a subnetwork of G of size greater than that of F by a factor logarithmic in all parameters of G except depth (possibly linear) that, without training, is a predictor that is with high probability close to F for all inputs. This paper improves on the the loss in the size of the network found from polynomial factor to a logarithmic. The reviewers found this an important paper studying the strong lottery ticket hypothesis. I recommend it for acceptance.